# Chronological Gene Expression of Human Gingival Fibroblasts with Low Reactive Level Laser (LLL) Irradiation

**DOI:** 10.3390/jcm10091952

**Published:** 2021-05-01

**Authors:** Yuki Wada, Asami Suzuki, Hitomi Ishiguro, Etsuko Murakashi, Yukihiro Numabe

**Affiliations:** 1Department of Periodontology, School of Life Dentistry at Tokyo, The Nippon Dental University, 1-9-20 Fujimi, Chiyoda-ku, Tokyo 102-8159, Japan; y-wada_d2117015@tky.ndu.ac.jp (Y.W.); hitomi-i@tky.ndu.ac.jp (H.I.); satoetsu@tky.ndu.ac.jp (E.M.); numabe-y@tky.ndu.ac.jp (Y.N.); 2Division of General Dentistry, The Nippon Dental University Hospital, 2-3-16 Fujimi, Chiyoda-ku, Tokyo 102-8158, Japan; 3Dental Education Support Center, School of Life Dentistry, The Nippon Dental University, 1-9-20 Fujimi, Chiyoda-ku, Tokyo 102-8159, Japan

**Keywords:** Low reactive Level Laser Therapy (LLLT), human gingival fibroblasts (HGF), microarray, differentially gene expression (DEGs), gene ontology, biological processes (BP), protein–protein interaction (PPI)

## Abstract

Though previously studies have reported that Low reactive Level Laser Therapy (LLLT) promotes wound healing, molecular level evidence was uncleared. The purpose of this study is to examine the temporal molecular processes of human immortalized gingival fibroblasts (HGF) by LLLT by the comprehensive analysis of gene expression. HGF was seeded, cultured for 24 h, and then irradiated with a Nd: YAG laser at 0.5 W for 30 s. After that, gene differential expression analysis and functional analysis were performed with DNA microarray at 1, 3, 6 and 12 h after the irradiation. The number of genes with up- and downregulated differentially expression genes (DEGs) compared to the nonirradiated group was large at 6 and 12 h after the irradiation. From the functional analysis results of DEGs, Biological Process (BP) based Gene Ontology (GO), BP ‘the defense response’ is considered to be an important process with DAVID. Additionally, the results of PPI analysis of DEGs involved in the defense response with STRING, we found that the upregulated DEGs such as CXCL8 and NFKB1, and the downregulated DEGs such as NFKBIA and STAT1 were correlated with multiple genes. We estimate that these genes are key genes on the defense response after LLLT.

## 1. Introduction

Periodontal disease is a chronic multifactorial inflammatory disease caused by genetic, immune, environmental, microbial factors and lifestyles, with anaerobic bacteria in the oral cavity as the main causative organism. Periodontal disease is generally treated with nonsurgical therapy, that is performed with hand or power-driven instrumentation. In recent years, combination therapies with scaler and laser, or laser alone have attracted attention [1,2,3].

The laser therapy methods currently used for treatment of periodontal diseases can be broadly divided into two types: High reactive Level Laser Therapy (HLLT) and Low reactive Level Laser Therapy (LLLT). 

HLLT is an application of laser intensity that produces an irreversible reaction (photobiological destruction reaction) beyond the cell survival region and is used for tissue incision and transpiration [4,5,6]. 

On the other hand, LLLT is a treatment that applies laser intensity to generate a reversible reaction (photobiologically active reaction) within the cell survival threshold. LLLT are expected to have anti-inflammatory effects [7,8,9,10,11,12]; pain relief [13], improvement/promotion of blood flow [14], activation of cells in tissues, wound healing by proliferation [15] and tissue regeneration without causing tissue degeneration with low-power laser irradiation conditions [16,17,18]. In recent years, the promotion of wound healing with LLLT has been one of the highlights.

Wound healing is thought to progress in the process of hemorrhagic coagulation phase, inflammatory phase, proliferative phase, reconstruction phase after injury by external stimulus. During the hemorrhagic coagulation phase, blood is coagulated by platelets, which is one of the coagulation factors, and growth factors such as platelet-derived growth factor (PDGF) and cytokines are released from the platelets. During the inflammatory phase, factors such as Nuclear Factor-κB (NF-κB) cause infiltration of inflammatory cells such as neutrophils and macrophages. Then, the release of growth factors and cytokines such as transforming growth factor-β (TGF-β) and fibroblast growth factor (FGF) is observed [19]. During the proliferative phase, it promotes the migration and proliferation of fibroblasts and keratinocytes [20]. Extracellular matrix is synthesized from fibroblasts and serves as a scaffold for cell migration and adhesion. During maturity, scar tissue formation occurs.

In the study of LLLT, TGF-β1 is closely involved in cell differentiation, migration, and adhesion by LLLT. In addition, it is thought to be involved in a wide range of areas such as ontogeny, tissue reconstruction, wound healing, inflammation/immunity, and cancer infiltration/metastasis. It has also been reported that the expression of NF-κB is increased [21]. 

Previous studies reported that the effects of low-reaction level laser irradiation on periodontium-derived cultured cells have been mainly on cell proliferation and cell transport ability related to wound healing. However, the elucidation of the mechanism at the molecular level leading to the promotion of wound healing by laser irradiation has been insufficient. It is considered that there may be a series of processes related to various wound healing by LLLT by the approach from biological processes (BP). There are few studies on mechanism analysis at the gene level using microarrays by LLLT for HGF, and only a limited number of studies have analyzed gene expression over time [22,23]. In order to clarify the effect of laser irradiation by LLLT, it is important to analyze and consider changes in gene expression and changes in BP of differentially expression genes (DEGs) over time in order to understand the mechanism at the molecular level. 

In this study, HGF was irradiated with LLLT, and gene expression fluctuations at 1, 3, 6, and 12 h after irradiation were analyzed using a DNA microarray. In addition, we focused on the defense response, which showed remarkable changes in gene expression over time in relation to wound healing obtained from the results of vast amounts of analytical data, and to investigate for the mechanism from the expression change genes and BP due to the photobiological effects of laser. The aim of this study was to elucidate the changes of gene expression on the wound healing, especially defense response, over time after irradiation.

## 2. Materials and Methods

### 2.1. Cell Culture 

Human immortalized gingival fibroblasts (HGF; Applied Biological Material, Richmond, BC, Canada) were used, and 10% Fetal Bovine Serum (Moregate, Bulimba, Australia), 50 U/mL Penicilin G, 50 μg/mL Amphotericin. The study was carried out by culturing in D-MEM/F-12 medium (Life Technologies Corporation, Grand Island, NY, USA) under 37 °C, and 5% CO_2_ conditions. The HGF at the time of irradiation was in the logarithmic growth phase. 

### 2.2. Dental Laser Device and Laser Irradiation Stent

A dental Nd: YAG laser: impulse dental laser (Incisive Japan Co., Ltd., Tokyo, Japan) was used as a dental laser device, and an ultrafine fiber with a diameter of 320 nm was used for the laser light guide tip. Stents (Gikousha, Kanagawa, Japan) were prepared to uniformly irradiate cells with laser, attached to a handpiece, and used for research. Irradiation conditions were found to be significantly different in cell proliferation curvature in previous studies, irradiation output conditions 0.5 W (100 mJ, 5 pps), irradiation time 30 s, irradiation distance from the tip of the fiber guide to each well plate. Laser irradiation was performed with a distance of 20 mm to the bottom [24].

### 2.3. Microarray Analysis

Total RNA was extracted from HGF with RNeasy^®^ Plus Micro Kit (QIAGEN, Valencia, CA, USA) before laser irradiation 1, 3, 6 and 12 h after irradiation on a 96-well plate. cDNA was synthesized from total RNA using the SuperScript™ VILO™ cDNA Synthesis Kit (Invitrogen, Carlsbad, CA, USA). After synthesis, the cDNA was fragmented and biotin labeled. Biotin-labeled cDNA was added to the GeneChipTM Human Gene 2.0 ST Array (Thermo Fisher Scientific Inc., Waltham, MA, USA) and hybridized with a probe (GeneChipTM Hybridization, Wash, and Stain Kit; Thermo Fisher Scientific Inc., Waltham, MA, USA). Phycoerythrin staining was performed, and the fluorescence signal was measured with a GeneChip scanner (Scanner 3000 7 G; Thermo Fisher Scientific Inc., Waltham, MA, USA). After normalization, Expression Gene was analyzed by SST-RMA algorithm.

### 2.4. Data Analysis of Differentially Expressed Genes (DEGs)

DEGs were extracted with Affymetrix ^®^ Expression ConsoleTM (Thermo Fisher Scientific Inc. Waltham, MA, USA). The cutoff values were fold change (FC) ≥ |1.5| and *p*-value < 0.05. The nonirradiated group (control), the irradiated group (test) were defined as upregulated DEGs with significantly increased expression, and downregulated DEGs with significantly decreased expression with respect to the control group.

### 2.5. Functional Analysis of Differentially Expressed Genes

A functional analysis of DEGs was performed based on Gene Ontology (GO) with the database for annotation, visualization and integrated discovery (DAVID). Enrichment analysis on the BP was performed on the DEGs at 1, 3, 6, and 12 h after irradiation. The cutoff value was a modified Fisher exact *p*-value < 0.1, total count ≤ 2. The DEGs contained in the upregulated and the downregulated regions at each irradiation time were analyzed.

### 2.6. Protein–Protein Interaction (PPI) of Up- or Downregulated DEGs

We focused on the defense response which is an important response on the wound healing process. PPI analysis of up- or downregulated DEGs, which were involved in defense response and continuously observed as up- or downregulated at 6 and 12 h, and related expression fluctuations were observed at all times after irradiation and Search Tool for the Retrieval of Interacting Genes (STRING) was performed. Furthermore, from the analysis of PPI, variation in the expression of genes correlated with other DEGs over time was analyzed.

## 3. Results

### 3.1. Extraction of DEGs

In order to compare the gene expression after laser irradiation on time course, the DEGs were extracted. DEGs were extracted under the condition that the cutoff value was FC ≥ |1.5| and *p*-value < 0.05. Control and test were defined as upregulated DEGs with significantly increased expression and downregulated DEGs with significantly decreased expression with respect to the control group. At 1 h after the irradiation, 83 upregulated and 50 downregulated genes were extracted (Appendix A). At 3 h after, 46 upregulated genes and 32 downregulated genes (Appendix A), at 6 h after, 362 upregulated and 549 downregulated genes (Appendix A) and at 12 h after, 253 upregulated genes and 413 downregulated were extracted (Appendix A).

The number of DEGs was large 6 and 12 h after the irradiation. At 6 h, the number of DEGs was the highest in both the upregulated gene and downregulated gene groups.

### 3.2. Functional Analysis on GO

From the results of functional analysis with DAVID, the number of BP related with each DEG after the irradiation was 13 BP on upregulated and 35 BP on downregulated DEGs at 1 h after, 6 BP on upregulated and 68 BP on downregulated DEGs at 3 h after, 212 BP on upregulated and 288 BP on downregulated DEGs at 6 h after, and 84 BP on upregulated and 425 BP on downregulated DEGs at 12 h after. The number of BP was small at 1 and 3 h, and the number of BP was large at 6 and 12 h. Table 1, Table 2, Table 3, Table 4, Table 5, Table 6, Table 7 and Table 8 show the top BPs for each irradiation time.

BPs on upregulated DEGs at 1 h after irradiation are, for example, GO: 0050867 ~ positive regulation of cell activation, GO: 0050865 ~ regulation of cell activation (Table 1), BPs on downregulated DEGs are, for example, GO: 0032774 ~ RNA biosynthetic process, GO: 0007267 ~ cell–cell signaling (Table 2). 

At 3 h after the irradiation, BPs on upregulated DEGs are, for example, GO: 0055085 ~ transmembrane transport, GO: 0006820 ~ anion transport (Table 3), BPs on down-]regulated DEGs are, for example, GO: 0032774 ~ RNA biosynthetic process, GO: 0016070 ~ RNA metabolic process (Table 4). 

At 6 h, BPs on upregulated DEGs are, for example, GO: 0008283 ~ cell proliferation, GO: 0007155 ~ cell adhesion GO: 0022610 ~ biological adhesion, GO: 0042127 ~ regulation of cell proliferation, GO: 0006952 ~ defense response, GO: 0060429 ~ epithelium development (Table 5), BP on downregulated DEGs are, for example, GO: 0034645 ~ cellular macromolecule biosynthetic process, GO: 0019438 ~ aromatic compound biosynthetic process, GO: 0006325 ~ chromatin organization, GO: 0007049 ~ cell cycle (Table 6).

At 12 h, BPs on upregulated DEGs are, for example, GO: 0006955 ~ immune response, GO: 0006952 ~ defense response, GO: 0009605 ~ response to external stimulus, GO: 0048584 ~ positive regulation of response to stimulus (Table 7), BPs on downregulated DEGs are, for example, GO: 0010468 BP such as ~ regulation of gene expression, GO: 0007155 ~ cell adhesion, GO: 0022610 ~ biological adhesion (Table 8).

### 3.3. PPI of Up- or Downregulated DEGs

As for the upregulated DEGs, there were DEGs involved in the defense response at all irradiation times. From the functional analysis of DAVID, we focused on the defense response related to wound healing, which is the BP of the DEGs of the upregulated DEGs at 6 and 12 h after the irradiation. Among the genes involved in the defense response, HCP5, DAPK3, IGLC2 and IGHV1 OR21-1 at 1 h after the irradiation for the upregulated DEGs, HP, IGHE and TPSAB1 at 3 h, CAKM2 B, IGHM, SEMA7 A, CXCL8 and TNFAIP6 at 6 h. IL34, ITGA2, SERPINE1, INHBA, PRDM1, LYZL4, PVR, BMP6, NFKB1, FOXP1, PER1, IGHG1, IGHA1, IGHA2, LDLR, SLC25 A6, PF4 and TGM2. At 12 h, it was IGHM, ITIH4, SEMA7 A, EDN1, HIST1 H2 BJ, KCNJ8, TNFAIP6, CCL21, CARD9, IFI6, SERPINB9, LYZL2, DEFB108 B, ECSIT, IGHG1, IGKC, IGHD, CHRFAM7 A and SLAMF6. 

As the results of PPI analysis of DEGs involved in the defense response with STRING, CXCL8 was a gene associated with multiple genes in the upregulated DEGs of the defense response. The relationships with SERPINE1, PF4, NFKB1, TNFAIP6, EDN1, CCL21, ITIH4 and HP were confirmed. In addition, HP, ITIH4, TNFAIP6 and NFKB1 were also associated with multiple genes (Figure 1). In the downregulated DEGs, STAT1 was a gene associated with multiple genes. In particular, it was associated with SMAD3, IL15, CCL22, NAIP, NRIH3, LY96, NFKBIA, PSMB8, PSMB9, PSMB10, OSA2, OAS3, IRF2, BCL6 and DDX58. In addition, NFKBIA, TLR3, IL15 and IRF2 were genes associated with multiple genes (Figure 2).

Figure 3 shows the changes in the expression of major genes related to other genes over time.

## 4. Discussion

Laser treatment has been studied for clinical application in many fields including medical and dentistry. In the field of dentistry, lasers are used for various purposes such as promoting wound healing, gingival incision, caries removal and sterilization/disinfection effect [25]. It has been reported that LLLT promotes wound healing and bone formation in the oral cavity by utilizing the bioactivating effect of chronic periodontitis.

In addition, at the cellular and tissue levels, there are research reports related to promotion of wound healing, such as cell proliferation of fibroblasts [15,20], osteoblasts [22,26] and vascular endothelial cells [18,27] by LLLT. However, clinical application in medical and dental treatments has not been carried out much. This is because the molecular biological findings are not clear.

Intracellular biological effects of LLLT include physiological activity by photoreceptors, changes in intracellular signal cascades, and changes in genes. The intracellular photoreceptor of LLLT is cytochrome c oxidase, which is a transmembrane protein complex that is an electron transport chain enzyme found in mitochondria. Promotion of cell proliferation by increasing cytochrome c oxidase activity and ATP is one of the representative mechanisms in LLLT research [28,29,30,31].

As a method for elucidating the mechanism of wound healing by LLLT, DNA microarrays are considered to be useful because they can examine thousands to tens of thousands of gene expressions at a time. From the obtained gene expression data, DEGs are extracted using bioinformatics analysis tools [32,33]. Furthermore, based on GO, by analyzing the biological process (BP) of DEGs, it is possible to estimate what is happening at each time by knowing the known functions of the gene contained in DEGs. It is also possible to infer what is about to happen at that time by performing chronological analysis. In addition, by searching for protein–protein interaction (PPI), it is possible to search for relationships at the molecular level [34,35]. The method plays a major role in elucidating the effects of proteins controlled by LLLT-stimulated genes at the molecular level.

In this study, we focused on the defense response from a huge amount of microarray data and analyzed the chronological changes in gene expression and the function of the genes after LLLT to HGF and the function of the genes.

The gene expression reactions related to wound healing with LLLT were remarkable 6–12 h after the irradiation. Analysis of the gene expression changes within these times were considered important for investigating the molecular mechanism of effects on HGF with LLLT. In particular, among the DEGs, those that are common over time and those with a significantly large expression fluctuation amount were considered largely affected by LLLT.

Analysis of BP over time revealed that upregulated were activated from 1 to 3 h after the initial irradiation, and downregulated BP was involved in RNA metabolism and activity. In both cases, the number of applicable DEGs for BP was 10 or less. It was suggested that there was no cohesive expression fluctuation as a function. 

At 6 h after the irradiation, upregulated DEGs were observed related to cell proliferation, adhesion and defense reaction. Additionally, downregulated DEGs were observed in many BPs involved in RNA metabolism, activity, cell polymer production, and metabolism.

At 12 h after the irradiation, upregulated DEGs were observed with many BPs involved in defense reaction, immune reaction and response to external stimuli, and downregulated DEGs were observed with many BPs involved in RNA metabolism, activity, cell polymer production and metabolism. In the upregulated group, many BPs associated with wound healing were observed at 6–12 h after irradiation. Additionally, in downregulated, similar BP such as RNA metabolism were observed from 1 to 12 h after irradiation.

The BP ‘the defense response’ focused on in this study belongs to the BP ‘the response to stress of response to stimulus’ in GO. The response to stimulus is the process by which the state or activity of a cell or organism changes as a result of stimulation. The response to stress also causes motility, secretion, enzyme production, gene expression, etc., as a result of impaired homeostasis of the organism or cell due to extrinsic factors (temperature, humidity, ionizing radiation). The defense response, which belongs to response to stress, is a reaction caused in response to the presence of foreign substances or the occurrence of injuries, and is an important BP involved in the restriction, prevention/recovery of damage to living organisms.

In the BP ‘defense response’, the protein encoded by CXCL8, which is a downregulated DEG, is called interleukin-8 (IL-8). IL-8 is secreted by mononuclear macrophages, neutrophils, eosinophils, T lymphocytes, epithelial cells, and fibroblasts. IL-8 is also known as a neutrophil chemotactic factor with two major functions. It induces chemotaxis of target cells to the infected site. IL-8 is also known as a strong promoter of angiogenesis. IL-8 expression is regulated by the transcription factor NF-κB [36,37,38,39,40,41,42].

In particular, the upregulated DEGs NFKB1 and the downregulated DEGs NFKBIA are genes involved in NF-κB. These are one of the genes involved in the existing mechanism. NFKB1 is a transcriptional regulator that is activated by various intracellular and extracellular stimuli such as cytokines, oxidant free radicals, UV irradiation and bacterial or viral products. Activated NFKB stimulates the expression of genes involved in biological functions associated with many biological processes such as inflammation, immunity, differentiation and cells [43]. NFKBIA is a member of a family of cellular proteins that function to inhibit NF-κB transcription factors and IκBα masks the nuclear localization signals (NLS) of NF-κB proteins and inactivates them in the cytoplasm. It inhibits NF-κB by isolating it into a state [44]. From this result, it can be seen that NFKB1 increases and NFKBIA decreases when irradiation is performed, so that the activity of the pathway containing NF-κB occurs. Additionally, the defense response of BP is considered to be strongly related to the NF-κB pathway.

In a study by Chen et al., NF-κB activity was observed 1–10 h after irradiation [21]. 

In this study as well, changes in the expression of NFKB1 and NFKBIA were observed 6 h after irradiation, and the results of analysis from the viewpoint of BP also suggest that the movement of NF-κB due to irradiation is an important process in the mechanism of LLLT. In this study, we focused on the defense response. Further, we also will need to focus on wound healing-related processes such as BP ‘the immune response’.

By adding analysis more time points, it is possible to analyze detailed time-series processes after LLL irradiation. As future developments, we plan to study other BPs, collect LLLT microarray data at different time points, and analyze the effects of laser irradiation on fibroblasts and molecular-level processes during the healing process. We will lead to the elucidation of molecular evidence in ‘the response to stress of response to stimulus’ by LLLT.

## 5. Conclusions

The time points of 1, 3, 6 and 12 h after LLL irradiation were compared over time. The most DEGs after the LLL irradiation on HGF were showed at 6 h upregulated gene. The number of DEGs peaked 6 h after irradiation and slightly decreased at 12 h after irradiation. From the time-dependent functional analysis, the upregulated DEGs were involved in BPs of cell proliferation, adhesion, and defense response related to wound healing from 6 h. In addition, defense response is one of the important mechanisms in BP after the irradiation. We found that the upregulated DEGs such as CXCL8 and NFKB1, and the downregulated DEGs such as NFKBIA and STAT1 were correlated with multiple genes from these PPI. From these results, irradiation of LLLT showed fluctuations in the expression of genes related to BP defense response.

## Figures and Tables

**Figure 1 jcm-10-01952-f001:**
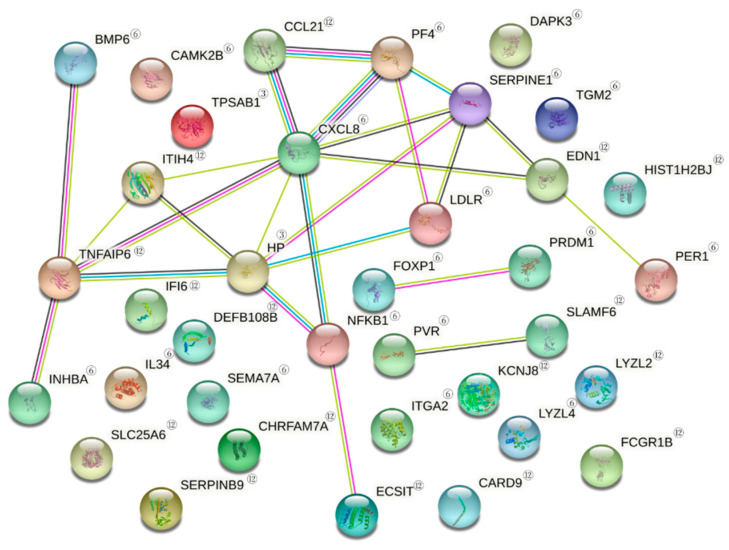
PPI of upregulated DEGs. Search tool STRING analysis of interacting genes and proteins reveals a protein–protein interaction PPI network in defense response by LLLT. PPI of upregulated DEGs related to ‘defense response’ Related genes after the irradiation at 3 h; ^③^, 6 h; ^⑥^ 12 h; ^⑫^ Red line: indicates the presence of fusion evidence; green line: neighborhood evidence; blue line: cooccurrence evidence; purple line: experimental evidence; yellow line: text mining evidence; light blue line: database evidence; black line: coexpression evidence.

**Figure 2 jcm-10-01952-f002:**
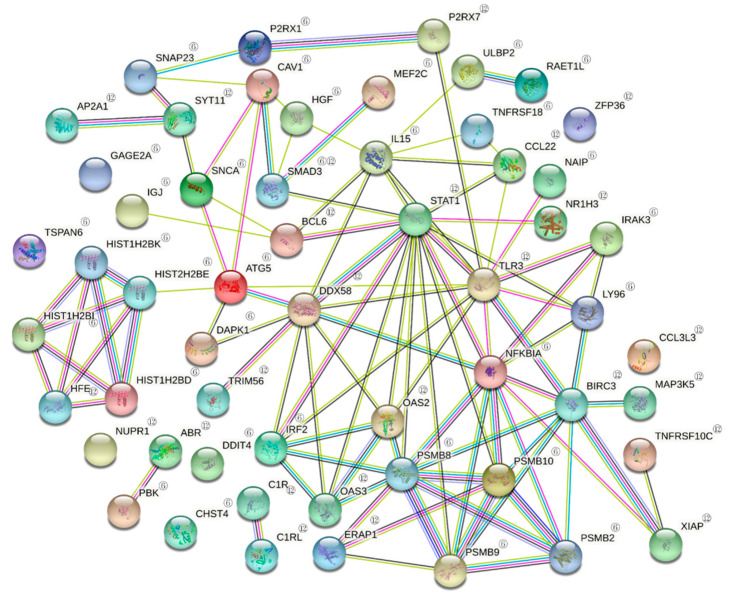
PPI of downregulated DEGs. Search tool STRING analysis of interacting genes and proteins reveals a protein–protein interaction network between proteins in defense response by LLLT. PPI of downregulated DEGs related to ‘defense response.’ Related genes after the irradiation at 3 h; ^③^, 6 h; ^⑥^ 12 h; ^⑫^.

**Figure 3 jcm-10-01952-f003:**
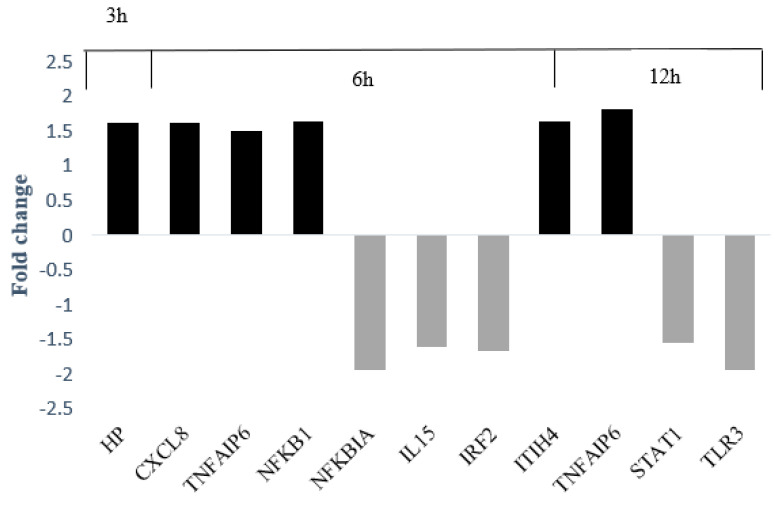
The FC of the key genes.

**Table 1 jcm-10-01952-t001:** The functional analysis of the upregulated genes at 1 h after Low Reactive Level Laser (LLL) irradiation.

Gene Ontology (GO) ID and Terms on Biological Process (BP)	Count	%	*p*-Value
GO:0061024~membrane organization	6	6.67	4.45 × 10^−2^
GO:0002696~positive regulation of leukocyte activation	4	4.44	2.15 × 10^−2^
GO:0050867~positive regulation of cell activation	4	4.44	2.31 × 10^−2^
GO:0002694~regulation of leukocyte activation	4	4.44	6.00 × 10^−2^
GO:0072657~protein localization to membrane	4	4.44	6.66 × 10^−2^
GO:0050865~regulation of cell activation	4	4.44	7.06 × 10^−2^
GO:0008037~cell recognition	3	3.33	3.60 × 10^−2^
GO:0072659~protein localization to plasma membrane	3	3.33	5.98 × 10^−2^
GO:1990778~protein localization to cell periphery	3	3.33	6.97 × 10^−2^
GO:0007009~plasma membrane organization	3	3.33	9.88 × 10^−2^
GO:0006910~phagocytosis, recognition	2	2.22	6.52 × 10^−2^
GO:2000243~positive regulation of reproductive process	2	2.22	8.66 × 10^−2^
GO:0006911~phagocytosis, engulfment	2	2.22	9.18 × 10^−2^

Count: genes involved in the term; percentage (%): involved genes/total genes; *p*-value: modified Fisher exact *p*-value.

**Table 2 jcm-10-01952-t002:** The functional analysis of the downregulated genes at 1 h after LLL irradiation.

Gene Ontology (GO) ID and Terms on Biological Process (BP)	Count	%	*p*-Value
GO:0032774~RNA biosynthetic process	10	18.52	1.27 × 10^−2^
GO:0034654~nucleobase-containing compound biosynthetic process	10	18.52	2.59 × 10^−2^
GO:0018130~heterocycle biosynthetic process	10	18.52	2.79 × 10^−2^
GO:0019438~aromatic compound biosynthetic process	10	18.52	2.85 × 10^−2^
GO:0016070~RNA metabolic process	10	18.52	3.87 × 10^−2^
GO:0034645~cellular macromolecule biosynthetic process	10	18.52	6.07 × 10^−2^
GO:0010467~gene expression	10	18.52	8.40 × 10^−2^
GO:0007267~cell–cell signaling	5	9.26	7.48 × 10^−2^
GO:0006614~SRP-dependent cotranslational protein targeting to membrane	3	5.56	4.74 × 10^−3^
GO:0006613~cotranslational protein targeting to membrane	3	5.56	5.44 × 10^−3^
GO:0045047~protein targeting to ER	3	5.56	5.54 × 10^−3^
GO:0072599~establishment of protein localization to endoplasmic reticulum	3	5.56	5.96 × 10^−3^
GO:0000184~nuclear-transcribed mRNA catabolic process, nonsense-mediated decay	3	5.56	7.67 × 10^−3^
GO:0070972~protein localization to endoplasmic reticulum	3	5.56	8.28 × 10^−3^
GO:0019083~viral transcription	3	5.56	1.60 × 10^−2^
GO:0006413~translational initiation	3	5.56	1.74 × 10^−2^
GO:0006612~protein targeting to membrane	3	5.56	1.76 × 10^−2^
GO:0019080~viral gene expression	3	5.56	1.79 × 10^−2^
GO:0000956~nuclear-transcribed mRNA catabolic process	3	5.56	2.05 × 10^−2^
GO:0044033~multiorganism metabolic process	3	5.56	2.20 × 10^−2^
GO:0006402~mRNA catabolic process	3	5.56	2.33 × 10^−2^
GO:0006401~RNA catabolic process	3	5.56	2.91 × 10^−2^
GO:0006364~rRNA processing	3	5.56	3.36 × 10^−2^
GO:0016072~rRNA metabolic process	3	5.56	3.52 × 10^−2^
GO:0042254~ribosome biogenesis	3	5.56	5.00 × 10^−2^
GO:0090150~establishment of protein localization to membrane	3	5.56	5.91 × 10^−2^
GO:0034655~nucleobase-containing compound catabolic process	3	5.56	6.25 × 10^−2^
GO:0034470~ncRNA processing	3	5.56	7.17 × 10^−2^
GO:0046700~heterocycle catabolic process	3	5.56	7.17 × 10^−2^
GO:0044270~cellular nitrogen compound catabolic process	3	5.56	7.38 × 10^−2^
GO:0019439~aromatic compound catabolic process	3	5.56	7.54 × 10^−2^
GO:1901361~organic cyclic compound catabolic process	3	5.56	8.23 × 10^−2^
GO:0019058~viral life cycle	3	5.56	8.55 × 10^−2^
GO:0022613~ribonucleoprotein complex biogenesis	3	5.56	9.37 × 10^−2^
GO:0072657~protein localization to membrane	3	5.56	9.67 × 10^−2^

Count: genes involved in the term; percentage (%): involved genes/total genes; *p*-value: modified Fisher exact *p*-value.

**Table 3 jcm-10-01952-t003:** The functional analysis of the upregulated genes at 3 h after LLL irradiation.

Gene Ontology (GO) ID and Terms on Biological Process (BP)	Count	%	*p*-Value
GO:0055085~transmembrane transport	5	0.30	5.87 × 10^−2^
GO:1901615~organic hydroxy compound metabolic process	3	0.18	7.85 × 10^−2^
GO:0006820~anion transport	3	0.18	9.85 × 10^−2^
GO:0051180~vitamin transport	2	0.12	3.81 × 10^−2^
GO:0006767~water-soluble vitamin metabolic process	2	0.12	9.58 × 10^−2^

Count: genes involved in the term; percentage (%): involved genes/total genes; *p*-value: modified Fisher exact *p*-value.

**Table 4 jcm-10-01952-t004:** The functional analysis of the downregulated genes at 3 h after LLL irradiation.

Gene Ontology (GO) ID and Terms on Biological Process (BP)	Count	%	*p*-Value
GO:0032774~RNA biosynthetic process	6	15.79	3.8 × 10^−2^
GO:0034654~nucleobase-containing compound biosynthetic process	6	15.79	5.85 × 10^−2^
GO:0018130~heterocycle biosynthetic process	6	15.79	6.14 × 10^−2^
GO:0019438~aromatic compound biosynthetic process	6	15.79	6.21 × 10^−2^
GO:0016070~RNA metabolic process	6	15.79	7.52 × 10^−2^
GO:0044085~cellular component biogenesis	5	13.16	5.62 × 10^−2^
GO:0006614~SRP-dependent cotranslational protein targeting to membrane	4	10.53	1.52 × 10^−5^
GO:0006613~cotranslational protein targeting to membrane	4	10.53	1.88 × 10^−5^
GO:0045047~protein targeting to ER	4	10.53	1.94 × 10^−5^
GO:0072599~establishment of protein localization to endoplasmic reticulum	4	10.53	2.17 × 10^−5^
GO:0000184~nuclear-transcribed mRNA catabolic process, nonsense-mediated decay	4	10.53	3.20 × 10^−5^
GO:0070972~protein localization to endoplasmic reticulum	4	10.53	3.60 × 10^−5^
GO:0019083~viral transcription	4	10.53	1.01 × 10^−4^
GO:0006413~translational initiation	4	10.53	1.15 × 10^−4^
GO:0006612~protein targeting to membrane	4	10.53	1.17 × 10^−4^
GO:0019080~viral gene expression	4	10.53	1.20 × 10^−4^
GO:0000956~nuclear-transcribed mRNA catabolic process	4	10.53	1.48 × 10^−4^
GO:0044033~multiorganism metabolic process	4	10.53	1.66 × 10^−4^
GO:0006402~mRNA catabolic process	4	10.53	1.83 × 10^−4^
GO:0006401~RNA catabolic process	4	10.53	2.60 × 10^−4^
GO:0006364~rRNA processing	4	10.53	3.26 × 10^−4^
GO:0016072~rRNA metabolic process	4	10.53	3.51 × 10^−4^
GO:0042254~ribosome biogenesis	4	10.53	6.20 × 10^−4^
GO:0090150~establishment of protein localization to membrane	4	10.53	8.15 × 10^−4^
GO:0034655~nucleobase-containing compound catabolic process	4	10.53	8.94 × 10^−4^
GO:0034470~ncRNA processing	4	10.53	1.12 × 10^−3^
GO:0046700~heterocycle catabolic process	4	10.53	1.12 × 10^−3^
GO:0044270~cellular nitrogen compound catabolic process	4	10.53	1.18 × 10^−3^
GO:0019439~aromatic compound catabolic process	4	10.53	1.22 × 10^−3^
GO:1901361~organic cyclic compound catabolic process	4	10.53	1.41 × 10^−3^
GO:0019058~viral life cycle	4	10.53	1.51 × 10^−3^
GO:0022613~ribonucleoprotein complex biogenesis	4	10.53	1.76 × 10^−3^
GO:0072657~protein localization to membrane	4	10.53	1.85 × 10^−3^
GO:0034660~ncRNA metabolic process	4	10.53	2.87 × 10^−3^
GO:0006412~translation	4	10.53	4.01 × 10^−3^
GO:0072594~establishment of protein localization to organelle	4	10.53	4.47 × 10^−3^
GO:0043043~peptide biosynthetic process	4	10.53	4.49 × 10^−3^
GO:0016071~mRNA metabolic process	4	10.53	4.58 × 10^−3^
GO:1902582~single-organism intracellular transport	4	10.53	5.04 × 10^−3^
GO:0006605~protein targeting	4	10.53	5.16 × 10^−3^
GO:0043604~amide biosynthetic process	4	10.53	5.92 × 10^−3^
GO:0006518~peptide metabolic process	4	10.53	7.94 × 10^−3^
GO:0044802~single-organism membrane organization	4	10.53	9.30 × 10^−3^
GO:0033365~protein localization to organelle	4	10.53	1.04 × 10^−2^
GO:0006396~RNA processing	4	10.53	1.07 × 10^−2^
GO:0016032~viral process	4	10.53	1.26 × 10^−2^
GO:0044764~multiorganism cellular process	4	10.53	1.28 × 10^−2^
GO:0044265~cellular macromolecule catabolic process	4	10.53	1.29 × 10^−2^
GO:0043603~cellular amide metabolic process	4	10.53	1.36 × 10^−2^
GO:0044403~symbiosis, encompassing mutualism through parasitism	4	10.53	1.37 × 10^−2^

Count: genes involved in the term; percentage (%): involved genes/total genes; *p*-value: modified Fisher exact *p*-value.

**Table 5 jcm-10-01952-t005:** The functional analysis of the upregulated genes at 6 h after LLL irradiation.

Gene Ontology (GO) ID and Terms on Biological Process (BP)	Count	%	*p*-Value
GO:0034645~cellular macromolecule biosynthetic process	72	19.20	4.50 × 10^−3^
GO:0010467~gene expression	67	17.87	9.63 × 10^−2^
GO:0016070~RNA metabolic process	65	17.33	1.49 × 10^−2^
GO:0010468~regulation of gene expression	64	17.07	4.90 × 10^−3^
GO:0051171~regulation of nitrogen compound metabolic process	64	17.07	6.75 × 10^−3^
GO:0019219~regulation of nucleobase-containing compound metabolic process	61	16.27	5.24 × 10^−3^
GO:0010556~regulation of macromolecule biosynthetic process	60	16.00	8.42 × 10^−3^
GO:2000112~regulation of cellular macromolecule biosynthetic process	59	15.73	7.09 × 10^−3^
GO:0034654~nucleobase-containing compound biosynthetic process	59	15.73	4.02 × 10^−2^
GO:0018130~heterocycle biosynthetic process	59	15.73	4.99 × 10^−2^
GO:0019438~aromatic compound biosynthetic process	59	15.73	5.24 × 10^−2^
GO:0051252~regulation of RNA metabolic process	58	15.47	3.04 × 10^−3^
GO:0097659~nucleic acid-templated transcription	57	15.20	5.26 × 10^−3^
GO:0032774~RNA biosynthetic process	57	15.20	1.10 × 10^−2^
GO:0006355~regulation of transcription, DNA-templated	54	14.40	8.34 × 10^−3^
GO:1903506~regulation of nucleic acid-templated transcription	54	14.40	9.42 × 10^−3^
GO:2001141~regulation of RNA biosynthetic process	54	14.40	1.04 × 10^−2^
GO:0006351~transcription, DNA-templated	53	14.13	1.32 × 10^−2^
GO:0010646~regulation of cell communication	47	12.53	7.96 × 10^−3^
GO:0023051~regulation of signaling	47	12.53	1.07 × 10^−2^
GO:0009893~positive regulation of metabolic process	46	12.27	1.53 × 10^−2^
GO:0010604~positive regulation of macromolecule metabolic process	43	11.47	2.02 × 10^−2^
GO:0009966~regulation of signal transduction	42	11.20	1.56 × 10^−2^
GO:0009892~negative regulation of metabolic process	40	10.67	1.19 × 10^−2^
GO:0007166~cell surface receptor signaling pathway	40	10.67	2.78 × 10^−2^
GO:0031325~positive regulation of cellular metabolic process	40	10.67	6.31 × 10^−2^
GO:0010605~negative regulation of macromolecule metabolic process	39	10.40	5.48 × 10^−3^
GO:0031324~negative regulation of cellular metabolic process	37	9.87	1.76 × 10^−2^
GO:0006366~transcription from RNA polymerase II promoter	35	9.33	1.02 × 10^−3^
GO:0006357~regulation of transcription from RNA polymerase II promoter	35	9.33	1.17 × 10^−3^
GO:0010628~positive regulation of gene expression	33	8.80	1.19 × 10^−3^
GO:0071310~cellular response to organic substance	32	8.53	6.76 × 10^−2^
GO:0051173~positive regulation of nitrogen compound metabolic process	31	8.27	8.31 × 10^−3^
GO:0048584~positive regulation of response to stimulus	31	8.27	4.62 × 10^−2^
GO:0009891~positive regulation of biosynthetic process	30	8.00	1.39 × 10^−2^
GO:0031328~positive regulation of cellular biosynthetic process	29	7.73	1.93 × 10^−2^
GO:0006468~protein phosphorylation	29	7.73	4.20 × 10^−2^
GO:0008283~cell proliferation	29	7.73	4.43 × 10^−2^
GO:0008219~cell death	29	7.73	7.43 × 10^−2^
GO:0010557~positive regulation of macromolecule biosynthetic process	28	7.47	1.29 × 10^−2^
GO:0045935~positive regulation of nucleobase-containing compound metabolic process	28	7.47	2.02 × 10^−2^
GO:0012501~programmed cell death	28	7.47	6.56 × 10^−2^
GO:0007155~cell adhesion	27	7.20	4.37 × 10^−2^
GO:0022610~biological adhesion	27	7.20	4.53 × 10^−2^
GO:2000026~regulation of multicellular organismal development	27	7.20	4.97 × 10^−2^
GO:0006915~apoptotic process	27	7.20	5.76 × 10^−2^
GO:0051254~positive regulation of RNA metabolic process	26	6.93	9.98 × 10^−3^
GO:1902531~regulation of intracellular signal transduction	26	6.93	7.47 × 10^−2^
GO:0048585~negative regulation of response to stimulus	25	6.67	1.26 × 10^−2^
GO:0009890~negative regulation of biosynthetic process	25	6.67	3.44 × 10^−2^
GO:0042127~regulation of cell proliferation	25	6.67	4.68 × 10^−2^
GO:0048646~anatomical structure formation involved in morphogenesis	24	6.40	3.17 × 10^−3^
GO:0010558~negative regulation of macromolecule biosynthetic process	24	6.40	3.25 × 10^−2^
GO:0031327~negative regulation of cellular biosynthetic process	24	6.40	4.89 × 10^−2^
GO:0010629~negative regulation of gene expression	24	6.40	5.04 × 10^−2^
GO:0045893~positive regulation of transcription, DNA-templated	23	6.13	3.43 × 10^−2^
GO:1903508~positive regulation of nucleic acid-templated transcription	23	6.13	3.43 × 10^−2^
GO:1902680~positive regulation of RNA biosynthetic process	23	6.13	3.94 × 10^−2^
GO:0002682~regulation of immune system process	23	6.13	4.19 × 10^−2^
GO:0006952~defense response	23	6.13	9.61 × 10^−2^
GO:0010648~negative regulation of cell communication	22	5.87	1.69 × 10^−2^
GO:0023057~negative regulation of signaling	22	5.87	1.74 × 10^−2^
GO:0080134~regulation of response to stress	22	5.87	3.76 × 10^−2^
GO:2000113~negative regulation of cellular macromolecule biosynthetic process	22	5.87	4.90 × 10^−2^
GO:0051240~positive regulation of multicellular organismal process	22	5.87	9.32 × 10^−2^
GO:0051094~positive regulation of developmental process	21	5.60	1.41 × 10^−2^

Count: genes involved in the term; percentage (%): involved genes/total genes; *p*-value: modified Fisher exact *p*-value.

**Table 6 jcm-10-01952-t006:** The functional analysis of the downregulated genes at 6 h after LLL irradiation.

Gene Ontology (GO) ID and Terms on Biological Process (BP)	Count	%	*p*-Value
GO:0034645~cellular macromolecule biosynthetic process	158	0.26	1.20 × 10^−6^
GO:0010467~gene expression	147	0.24	2.91 × 10^−3^
GO:0019438~aromatic compound biosynthetic process	141	0.23	4.17 × 10^−6^
GO:0018130~heterocycle biosynthetic process	138	0.23	1.59 × 10^−5^
GO:0034654~nucleobase-containing compound biosynthetic process	135	0.22	3.51 × 10^−5^
GO:0016070~RNA metabolic process	132	0.22	2.23 × 10^−3^
GO:0051171~regulation of nitrogen compound metabolic process	128	0.21	1.10 × 10^−3^
GO:0010468~regulation of gene expression	126	0.21	1.36 × 10^−3^
GO:0032774~RNA biosynthetic process	121	0.20	1.21 × 10^−4^
GO:2000112~regulation of cellular macromolecule biosynthetic process	121	0.20	3.10 × 10^−4^
GO:0010556~regulation of macromolecule biosynthetic process	121	0.20	9.90 × 10^−4^
GO:0019219~regulation of nucleobase-containing compound metabolic process	118	0.20	2.92 × 10^−3^
GO:0097659~nucleic acid-templated transcription	113	0.19	8.72 × 10^−4^
GO:0006351~transcription, DNA-templated	109	0.18	8.23 × 10^−4^
GO:0006355~regulation of transcription, DNA-templated	107	0.18	1.77 × 10^−3^
GO:1903506~regulation of nucleic acid-templated transcription	107	0.18	2.18 × 10^−3^
GO:2001141~regulation of RNA biosynthetic process	107	0.18	2.60 × 10^−3^
GO:0051252~regulation of RNA metabolic process	107	0.18	7.16 × 10^−3^
GO:0009892~negative regulation of metabolic process	90	0.15	9.30 × 10^−6^
GO:0010605~negative regulation of macromolecule metabolic process	86	0.14	4.06 × 10^−6^
GO:0031324~negative regulation of cellular metabolic process	85	0.14	1.02 × 10^−5^
GO:0044085~cellular component biogenesis	85	0.14	1.03 × 10^−2^
GO:0043933~macromolecular complex subunit organization	83	0.14	1.72 × 10^−4^
GO:0009893~positive regulation of metabolic process	83	0.14	5.25 × 10^−2^
GO:0010604~positive regulation of macromolecule metabolic process	81	0.13	2.37 × 10^−2^
GO:0022607~cellular component assembly	79	0.13	6.13 × 10^−3^
GO:0051276~chromosome organization	67	0.11	7.48.E-12
GO:0033554~cellular response to stress	66	0.11	6.62 × 10^−5^
GO:0032268~regulation of cellular protein metabolic process	65	0.11	6.56 × 10^−2^
GO:0031327~negative regulation of cellular biosynthetic process	63	0.10	1.62 × 10^−6^
GO:0009890~negative regulation of biosynthetic process	63	0.10	2.75 × 10^−6^
GO:0051172~negative regulation of nitrogen compound metabolic process	63	0.10	2.91 × 10^−6^
GO:0010558~negative regulation of macromolecule biosynthetic process	61	0.10	1.79 × 10^−6^
GO:0010629~negative regulation of gene expression	61	0.10	7.71 × 10^−6^
GO:2000113~negative regulation of cellular macromolecule biosynthetic process	60	0.10	4.10 × 10^−7^
GO:0071822~protein complex subunit organization	60	0.10	4.18 × 10^−4^
GO:0045934~negative regulation of nucleobase-containing compound metabolic process	59	0.10	2.74 × 10^−6^
GO:0065003~macromolecular complex assembly	59	0.10	1.21 × 10^−3^
GO:0006461~protein complex assembly	56	0.09	9.86 × 10^−5^
GO:0070271~protein complex biogenesis	56	0.09	1.00 × 10^−4^
GO:0007049~cell cycle	56	0.09	3.16 × 10^−3^
GO:0006325~chromatin organization	54	0.09	4.01 × 10^−13^
GO:1903507~negative regulation of nucleic acid-templated transcription	53	0.09	2.84 × 10^−6^
GO:1902679~negative regulation of RNA biosynthetic process	53	0.09	4.28 × 10^−6^
GO:0051253~negative regulation of RNA metabolic process	53	0.09	1.24 × 10^−5^
GO:0045892~negative regulation of transcription, DNA-templated	52	0.09	1.93 × 10^−6^
GO:0034622~cellular macromolecular complex assembly	47	0.08	4.38 × 10^−6^
GO:0031399~regulation of protein modification process	47	0.08	9.08 × 10^−2^
GO:0044248~cellular catabolic process	46	0.08	8.76 × 10^−2^
GO:0022402~cell cycle process	44	0.07	1.65 × 10^−2^

Count: genes involved in the term; percentage (%): involved genes/total genes; *p*-value: modified Fisher exact *p*-value.

**Table 7 jcm-10-01952-t007:** The functional analysis of the upregulated genes at 12 h after LLL irradiation.

Gene Ontology (GO) ID and Terms on Biological Process (BP)	Count	%	*p*-Value
GO:0007166~cell surface receptor signaling pathway	28	10.73	2.37 × 10^−2^
GO:0006955~immune response	24	9.20	3.39 × 10^−4^
GO:0006952~defense response	23	8.81	6.37 × 10^−4^
GO:0009605~response to external stimulus	23	8.81	2.57 × 10^−2^
GO:0048584~positive regulation of response to stimulus	22	8.43	3.73 × 10^−2^
GO:0003008~system process	21	8.05	5.96 × 10^−2^
GO:0002682~regulation of immune system process	19	7.28	6.30 × 10^−3^
GO:0050776~regulation of immune response	18	6.90	1.70 × 10^−4^
GO:0016192~vesicle-mediated transport	17	6.51	5.14 × 10^−2^
GO:0045087~innate immune response	16	6.13	6.31 × 10^−4^
GO:0007186~G-protein coupled receptor signaling pathway	16	6.13	2.69 × 10^−2^
GO:0051707~response to other organism	15	5.75	1.72 × 10^−3^
GO:0043207~response to external biotic stimulus	15	5.75	1.72 × 10^−3^
GO:0009607~response to biotic stimulus	15	5.75	2.76 × 10^−3^
GO:0050877~neurological system process	15	5.75	6.02 × 10^−2^
GO:0006897~endocytosis	14	5.36	6.77 × 10^−4^
GO:0002252~immune effector process	14	5.36	1.83 × 10^−3^
GO:0001775~cell activation	14	5.36	1.00 × 10^−2^
GO:0002684~positive regulation of immune system process	14	5.36	1.51 × 10^−2^
GO:0009617~response to bacterium	13	4.98	3.80 × 10^−4^
GO:0050778~positive regulation of immune response	13	4.98	2.71 × 10^−3^
GO:0045321~leukocyte activation	13	4.98	5.47 × 10^−3^
GO:0051094~positive regulation of developmental process	13	4.98	7.29 × 10^−2^
GO:0002768~immune response-regulating cell surface receptor signaling pathway	12	4.60	9.96 × 10^−5^
GO:0002764~immune response-regulating signaling pathway	12	4.60	1.12 × 10^−3^
GO:0046649~lymphocyte activation	12	4.60	4.91 × 10^−3^
GO:0002449~lymphocyte mediated immunity	11	4.21	2.83 × 10^−5^
GO:0002443~leukocyte mediated immunity	11	4.21	1.79 × 10^−4^
GO:0002250~adaptive immune response	11	4.21	6.62 × 10^−4^
GO:0098542~defense response to other organism	11	4.21	2.23 × 10^−3^
GO:0042742~defense response to bacterium	10	3.83	9.53 × 10^−5^
GO:0002460~adaptive immune response based on somatic recombination of immune receptors built from immunoglobulin superfamily domains	10	3.83	1.65 × 10^−4^
GO:0002429~immune response-activating cell surface receptor signaling pathway	10	3.83	1.01 × 10^−3^
GO:0002757~immune response-activating signal transduction	10	3.83	7.63 × 10^−3^
GO:0002253~activation of immune response	10	3.83	1.40 × 10^−2^
GO:0016064~immunoglobulin mediated immune response	9	3.45	2.44 × 10^−5^
GO:0019724~B cell mediated immunity	9	3.45	2.66 × 10^−5^
GO:0042113~B cell activation	9	3.45	3.08 × 10^−4^
GO:0006959~humoral immune response	9	3.45	3.25 × 10^−4^
GO:0051251~positive regulation of lymphocyte activation	9	3.45	8.51 × 10^−4^
GO:0002696~positive regulation of leukocyte activation	9	3.45	1.42 × 10^−3^
GO:0050867~positive regulation of cell activation	9	3.45	1.70 × 10^−3^
GO:0051249~regulation of lymphocyte activation	9	3.45	7.68 × 10^−3^
GO:0002694~regulation of leukocyte activation	9	3.45	1.59 × 10^−2^
GO:0050865~regulation of cell activation	9	3.45	2.28 × 10^−2^
GO:0006958~complement activation, classical pathway	8	3.07	5.50 × 10^−6^
GO:0002455~humoral immune response mediated by circulating immunoglobulin	8	3.07	1.24 × 10^−5^
GO:0006956~complement activation	8	3.07	1.65 × 10^−5^
GO:0072376~protein activation cascade	8	3.07	6.86 × 10^−5^
GO:0006909~phagocytosis	8	3.07	3.11 × 10^−3^

Count: genes involved in the term; percentage (%): involved genes/total genes; *p*-value: modified Fisher exact *p*-value.

**Table 8 jcm-10-01952-t008:** The functional analysis of the downregulated genes at 12 h after LLL irradiation.

Gene Ontology (GO) ID and Terms on Biological Process (BP)	Count	%	*p*-Value
GO:0034645~cellular macromolecule biosynthetic process	102	23.83	6.82 × 10^−2^
GO:0051171~regulation of nitrogen compound metabolic process	94	21.96	2.85 × 10^−2^
GO:0010556~regulation of macromolecule biosynthetic process	90	21.03	1.75 × 10^−2^
GO:0010468~regulation of gene expression	90	21.03	6.33 × 10^−2^
GO:0018130~heterocycle biosynthetic process	90	21.03	8.55 × 10^−2^
GO:0034654~nucleobase-containing compound biosynthetic process	89	20.79	8.55 × 10^−2^
GO:0019219~regulation of nucleobase-containing compound metabolic process	88	20.56	3.19 × 10^−2^
GO:2000112~regulation of cellular macromolecule biosynthetic process	87	20.33	2.26 × 10^−2^
GO:0097659~nucleic acid-templated transcription	82	19.16	2.93 × 10^−2^
GO:0032774~RNA biosynthetic process	82	19.16	6.02 × 10^−2^
GO:0051252~regulation of RNA metabolic process	81	18.93	3.76 × 10^−2^
GO:0006351~transcription, DNA-templated	80	18.69	2.03 × 10^−2^
GO:1903506~regulation of nucleic acid-templated transcription	80	18.69	2.30 × 10^−2^
GO:2001141~regulation of RNA biosynthetic process	80	18.69	2.58 × 10^−2^
GO:0006355~regulation of transcription, DNA-templated	79	18.46	2.76 × 10^−2^
GO:0023051~regulation of signaling	77	17.99	1.04 × 10^−3^
GO:0010646~regulation of cell communication	76	17.76	1.04 × 10^−3^
GO:0009966~regulation of signal transduction	71	16.59	6.57 × 10^−4^
GO:0065009~regulation of molecular function	71	16.59	2.42 × 10^−3^
GO:0035556~intracellular signal transduction	69	16.12	8.62 × 10^−4^
GO:0009893~positive regulation of metabolic process	68	15.89	3.21 × 10^−2^
GO:0010604~positive regulation of macromolecule metabolic process	67	15.65	1.20 × 10^−2^
GO:0006796~phosphate-containing compound metabolic process	67	15.65	5.25 × 10^−2^
GO:0006793~phosphorus metabolic process	67	15.65	5.42 × 10^−2^
GO:0031325~positive regulation of cellular metabolic process	66	15.42	1.53 × 10^−2^
GO:0044085~cellular component biogenesis	65	15.19	3.58 × 10^−2^
GO:0050790~regulation of catalytic activity	63	14.72	7.74 × 10^−4^
GO:0022607~cellular component assembly	63	14.72	9.06 × 10^−3^
GO:0007166~cell surface receptor signaling pathway	59	13.79	5.38 × 10^−2^
GO:0051128~regulation of cellular component organization	57	13.32	9.42 × 10^−3^
GO:0016310~phosphorylation	56	13.08	6.37 × 10^−3^
GO:0051246~regulation of protein metabolic process	55	12.85	7.26 × 10^−2^
GO:0033554~cellular response to stress	53	12.38	1.80 × 10^−4^
GO:0007399~nervous system development	52	12.15	2.06 × 10^−2^
GO:0031324~negative regulation of cellular metabolic process	52	12.15	6.82 × 10^−2^
GO:0032268~regulation of cellular protein metabolic process	51	11.92	9.24 × 10^−2^
GO:1902531~regulation of intracellular signal transduction	50	11.68	5.12 × 10^−4^
GO:0044093~positive regulation of molecular function	50	11.68	2.35 × 10^−3^
GO:0006928~movement of cell or subcellular component	48	11.21	4.02 × 10^−3^
GO:0009605~response to external stimulus	47	10.98	6.26 × 10^−2^
GO:0043085~positive regulation of catalytic activity	46	10.75	7.29 × 10^−4^
GO:0007049~cell cycle	46	10.75	2.94 × 10^−3^
GO:0006357~regulation of transcription from RNA polymerase II promoter	46	10.75	1.51 × 10^−2^
GO:1902589~single-organism organelle organization	45	10.51	2.52 × 10^−3^
GO:0006366~transcription from RNA polymerase II promoter	45	10.51	2.03 × 10^−2^
GO:0008219~cell death	44	10.28	8.02 × 10^−2^
GO:0031399~regulation of protein modification process	43	10.05	1.20 × 10^−2^
GO:0048585~negative regulation of response to stimulus	41	9.58	1.17 × 10^−3^
GO:0007155~cell adhesion	41	9.58	3.80 × 10^−2^
GO:0022610~biological adhesion	41	9.58	3.98 × 10^−2^

Count: genes involved in the term; percentage (%): involved genes/total genes; *p*-value: modified Fisher exact *p*-value.

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
