# Peer review of "Chronological Gene Expression of Human Gingival Fibroblasts with Low Reactive Level Laser (LLL) Irradiation"

_jcm, 2021, doi:10.3390/jcm10091952_

Round 1

Reviewer 1 Report

Dear authors,

after revision, I considered enough all information provided.

Congratulations.

Reviewer 2 Report

The authors received the suggestions of the reviewers and applied them in text.

Reviewer 3 Report

The authors have revised the manuscript appropriately in response to the reviewers' comments. I think this manuscript is suitable for publication. 

This manuscript is a resubmission of an earlier submission. The following is a list of the peer review reports and author responses from that submission.

Round 1

Reviewer 1 Report

P.14 L342-355: Please correct to the appropriate information.

Institutional Review Board Statement, Informed Consent Statement, and Data availability Statement are incorrect. So, please correct to the appropriate information.

Author Response

Thank you for your comments

Institutional Review Board Statement, Informed Consent Statement, and Data availability Statement are incorrect. So, please correct to the appropriate information.

Since this study uses purchased immortalized cells, it has not been certified by the Ethics Committee or consented to informed consent.

Institutional Review Board Statement: “Not applicable”

Informed Consent Statement: “Not applicable”

Data Availability Statement: Please refer to suggested Data Availability Statements in section “MDPI Research Data Policies” at https://www.mdpi.com/ethics.

Reviewer 2 Report

The manuscript is an interesting in vitro study focused on the genetic effect of low level laset irradiation conducted on human gingival fibroblast. The study is generally well conducted, but some important issue should be clarified before the manuscript can be further considered:

  • English language needs to be improved. I suggest to seek the expertize of a professional copy-editor, to correct grammar and the meaning of some sentences.
  • Page 1, line 31 "Conventionally...." The meaning of this sentence is not clear, the authors should rephrase as follow: " Periodontal disease is generally treated with non surgical therapy, that is performed with hand or power-driven instrumentation."
  • Page 12, line 232. In the discussion section the authors should compare the results with those achieved by similar studies on that topic.

Author Response

Thank you for your comments

  1. English language needs to be improved. I suggest to seek the expertize of a professional copy-editor, to correct grammar and the meaning of some sentences.

→After review report revision, we use MDPI's English editing service.

  1. Page 1, line 31 "Conventionally...." The meaning of this sentence is not clear, the authors should rephrase as follow: " Periodontal disease is generally treated with non surgical therapy, that is performed with hand or power-driven instrumentation."

We have corrected the text.

Conventionally, these have been treated using a hand scaler or an ultrasonic scaler. →Periodontal disease is generally treated with nonsurgical therapy, that is performed with hand or power-driven instrumentation.

  1. Page 12, line 232. In the discussion section the authors should compare the results with those achieved by similar studies on that topic.

→Genes related to references 21 have been studied for NFKB1 and NFKBIA. Other key genes have not yet been reported.

Reviewer 3 Report

Congratulations on the development of the study"Chronological Gene Expression of Human Gingival Fibroblasts with Low Reactive Level Laser (LLL) Irradiation"

I have some concerns about the study, which are described below.

  1. Was this study previously submitted to the ethics committee?
  2. There is bias on the laser application with a tip of 320 nm of diameter. Could the authors explain me better about it.
  3. Lines 97-98: "irradiation output conditions 0.5 W (100mJ, 5pps), irradiation time 30 seconds". Where these standardizations used were based from? and Why them?
  4. Where are the tables for the control group? This lack of information becomes the article non-published.
  5. The control group seemed to be forgotten. Clarify control group' data.
  6. The authors should have carefulness to affirm what was inserted in the conclusion, once there are limitations in the study.

Author Response

Thank you for your comments

  1. Was this study previously submitted to the ethics committee?

Since this study uses purchased immortalized cells, it has not been certified by the Ethics Committee or consented to informed consent.

  1. There is bias on the laser application with a tip of 320 nm of diameter. Could the authors explain me better about it.

We have corrected the text.

There was a mistake in the description. Changed from 320nm to 320μm.

  1. Lines 97-98: "irradiation output conditions 0.5 W (100mJ, 5pps), irradiation time 30 seconds". Where these standardizations used were based from? and Why them?

It was determined based on the standards and irradiation conditions from the 20 and 24 studies in the reference list.

  1. Where are the tables for the control group? This lack of information becomes the article non-published.
  2. The control group seemed to be forgotten. Clarify control group' data.

→These data are a comparison of the irradiated group (test) with the non-irradiated group (control) as the reference.

The up-regulated gene at each irradiation time means that the gene expression is higher in the test group than in the control group. Conversely, down-regulated gene means that gene expression is lower in the test group than in the control group.

In other words, the up-regulated gene is large in the test group and small in the control group, and the down-regulated gene is large in the control group and small in the test group.

  1. The authors should have carefulness to affirm what was inserted in the conclusion, once there are limitations in the study.

→We have corrected the text. We have rewritten the conclusion.

The time points of 1, 3, 6 and 12 hours after LLL irradiation were compared over time. The most DEGs after the LLL irradiation on HGF were showed at 6 hour up-regulated gene.  The number of DEGs peaked 6 hours after irradiation and slightly decreased at 12 hours after irradiation. From the time-dependent functional analysis, the up-regulated DEGs gene were involved in BPs of cell proliferation, adhesion, and defense response related to wound healing from 6 hours. In addition, defense response is one of the important mechanisms in BP after the irradiation. We found that the up-regulated DEGs such as CXCL8 and NFKB1, and the down-regulated DEGs such as NFKBIA and STAT1 were correlated with multiple genes from these PPI. From these results, irradiation of LLLT showed fluctuations in the expression of genes related to BP defense response.
